# Neurotoxins Acting on TRPV1—Building a Molecular Template for the Study of Pain and Thermal Dysfunctions

**DOI:** 10.3390/toxins17020064

**Published:** 2025-01-31

**Authors:** Florian Beignon, Margaux Notais, Sylvie Diochot, Anne Baron, Ziad Fajloun, Hélène Tricoire-Leignel, Guy Lenaers, César Mattei

**Affiliations:** 1University of Angers, INSERM U1083, CNRS UMR6015, MITOVASC, SFR ICAT, F-49000 Angers, France; florian.beignon@univ-angers.fr (F.B.); margaux_notais@hotmail.fr (M.N.); helene.tricoire-leignel@univ-angers.fr (H.T.-L.); guy.lenaers@univ-angers.fr (G.L.); 2Université Côte d’Azur, CNRS U7275, INSERM U1323, IPMC (Institut de Pharmacologie Moléculaire et Cellulaire), LabEx ICST (Laboratory of Excellence in Ion Channel Science and Therapeutics), FHU InovPain (Fédération Hospitalo-Universitaire “Innovative Solutions in Refractory Chronic Pain”), 660 Route des Lucioles, Sophia-Antipolis, F-06560 Nice, France; diochot@ipmc.cnrs.fr (S.D.); anne.baron@ipmc.cnrs.fr (A.B.); 3Laboratory of Applied Biotechnology (LBA3B), Department of Cell Culture, Azm Center for Research in Biotechnology and Its Applications, EDST, Lebanese University, Tripoli 1300, Lebanon; zfajloun@gmail.com; 4Service de Neurologie, CHU d’Angers, F-49000 Angers, France

**Keywords:** TRPV1, pain, thermoregulation, toxins, venoms, agonists, antagonists

## Abstract

Transient Receptor Potential (TRP) channels are ubiquitous proteins involved in a wide range of physiological functions. Some of them are expressed in nociceptors and play a major role in the transduction of painful stimuli of mechanical, thermal, or chemical origin. They have been described in both human and rodent systems. Among them, TRPV1 is a polymodal channel permeable to cations, with a highly conserved sequence throughout species and a homotetrameric structure. It is sensitive to temperature above 43 °C and to pH below 6 and involved in various functions such as thermoregulation, metabolism, and inflammatory pain. Several *TRPV1* mutations have been associated with human channelopathies related to pain sensitivity or thermoregulation. TRPV1 is expressed in a large part of the peripheral and central nervous system, most notably in sensory C and Aδ fibers innervating the skin and internal organs. In this review, we discuss how the transduction of nociceptive messages is activated or impaired by natural compounds and peptides targeting TRPV1. From a pharmacological point of view, capsaicin—the spicy ingredient of chilli pepper—was the first agonist described to activate TRPV1, followed by numerous other natural molecules such as neurotoxins present in plants, microorganisms, and venomous animals. Paralleling their adaptive protective benefit and allowing venomous species to cause acute pain to repel or neutralize opponents, these toxins are very useful for characterizing sensory functions. They also provide crucial tools for understanding TRPV1 functions from a structural and pharmacological point of view as this channel has emerged as a potential therapeutic target in pain management. Therefore, the pharmacological characterization of TRPV1 using natural toxins is of key importance in the field of pain physiology and thermal regulation.

## 1. Introduction

Pain is defined as “an unpleasant sensory and emotional experience associated with or resembling that associated with actual or potential tissue damage” [1]. The ability to detect painful stimuli is essential for species survival and well-being because it avoids mobilizing an injured tissue and compromising its long-term functioning. Nociception includes all the physiological processes integrating a painful stimulus to signal potential or a real tissue damage. Therefore, perception of pain relies on the sensitivity from an altered tissue, which is projected towards central nervous structures before being integrated into the emotional and memory networks [2]. There are three types of nociceptive stimuli: mechanical, chemical, and thermal. One stimulation is defined as harmful if it activates sensory receptors with a very high activation threshold [2]. The nociceptive signal is initiated by nociceptors, i.e., sensory neurons that convey a pain sensation to the central nervous system. These nociceptors are usually Aδ or C fibers, which are weakly myelinated or unmyelinated, respectively [3]. Pain signal transduction is therefore initiated at nociceptor peripheral nerve endings through a membrane depolarization called receptor potential. This depolarization generates action potentials, which propagate from the peripheral terminals to the spinal cord. Then it is transmitted and integrated into the brain by the spino-thalamic axis [4,5].

The TRP (Transient Receptor Potential) channel family gathers throughout species receptors mainly permeable to Ca^2+^ and Na^+^ ions [6] and is divided into six subfamilies: TRPA (ankyrin), TRPM (melastatin), TRPV (vanilloid), TRPC (canonical), TRPP (polycystin), and TRPML (mucolipin) [4,5]. Some of them are expressed in nociceptors and thus involved in the transduction of a nociceptive stimulus. Their activation in the sensory afferences induces membrane depolarization, which ultimately generates a painful stimulation in the central nervous system [7].

### 1.1. TRPV1: Structure and Function

The Transient Receptor Potential Vanilloid 1 (TRPV1) channel was the first member identified in the TRPV subfamily (Figure 1A,B). In this review, “TRPV1” will be used to describe common properties of the hTRPV1 or the rTRPV1 subunit, and each will be only specified when needed. TRPV1 was initially called the capsaicin (CAP) receptor or vanilloid receptor 1 (VR1) because of its affinity for these pungent compounds which activate and open the channel allowing Ca^2+^ and Na^+^ influx [8]. *VR1* cloning in rat identified a sequence similar to other TRP channels [9], which allowed the authors to rename this channel TRPV1, according to the current nomenclature. CAP is a plant alkaloid present in chili pepper (*Capsicum* spp.) responsible for the irritating, even burning, sensation induced while eating a spicy dish (Figure 1C). CAP contributed to the structural and functional characterization of TRPV1 [10,11].

TRPV1 is a polymodal receptor detecting chemical and thermal signals [12] and is expressed in C and Aδ fibers, but also in non-neuronal and non-nociceptive cells. As a heat thermosensor, TRPV1 is activated by temperatures above 43 °C and below 52 °C [9,13,14]. It is also activated by pH decrease, during acidification of the extracellular environment (pH < 6.0) or alkalinization of the intracellular medium [15]. High-temperature and acidification solicitations of mammalian TRPV1 are characteristic of tissue injury and inflammation and promote neuronal excitability. These two stimuli also potentiate one another: combining two subliminar heat and acidic stimuli can lead to TRPV1 activation. This is also the case with the third mechanical stimulus. This polymodal activation process of TRPV1, able to integrate several stimuli, supports its sensitization in inflammatory/pathological conditions.

Clinical data evidence the involvement of TRPV1 in cutaneous pain in humans. A severe acidification (pH = 5.0) of the skin was associated with non-adapting pain that can be inhibited by capsazepine (CPZ)—the main TRPV1 antagonist—whereas moderate acidic (pH = 6.0) pain was attributed to ASIC (Acid-Sensing Ion Channel) activation [16,17]. A clinical psychophysical study on 32 healthy volunteers suggested that TRPV1 is the predominant sensor of acidic pH-induced pain on skin [18], and two studies with cultured human DRG neurons, obtained from therapeutic ganglionectomy on patients suffering from chronic intractable pain, showed sustained acid- and CAP-activated, CPZ-inhibited TRPV1-like currents resulting in prolonged depolarizations and action potential firings [19,20]. The rTRPV1 subunit contains 838 amino acids (839 in hTRPV1) and forms homotetramers. Like other TRP channel subunits, TRPV1 contains six transmembrane segments (S1 to S6); three extracellular loops, including the third one between S5 and S6 forming the channel pore; and two intracellular loops (Figure 1A,B). TRPV1 also shares several topological and structural features with voltage-gated K^+^ channels. Both N- and C-terminal ends are in the intracellular compartment, and an Ankyrin Repeated Domain (ARD) has been characterized in the N-terminal domain [6,12,21]. TRPV1’s three-dimensional structure, achieved by cryo-electron microscopy, revealed the channel dimensions (100 Å long × 110 Å wide × 110 Å high) [22] and key binding sites, such as the vanilloid pockets. For instance, a first vanilloid pocket was identified in-between the interaction site formed by S3 + S4 and the intracellular loop between S4 and S5 [11,23], thus shedding light on TRPV1 activation mechanisms in the presence of its agonists [24,25,26]. The channel tetramerization is highly dependent on the TRP domain in the C-terminal segment close to the pore region [27].

### 1.2. TRPV1 Modulation

The regulation of TRPV1 activity is widely achieved by sensitization mechanisms. Indeed, TRPV1 can be sensitized by agonists or certain lipids, as well as by extracellular environment parameters such as temperature and acidity. TRPV1 sensitization by the environment’s acidification occurs through acidic residues located on TRPV1 extracellular domains, especially at the transmembrane segments S5 and S6 and at the pore-forming helix (E600, T633, F660), as well as at the loop between segments S3 and S4 (V538) [28,29,30].

ATP can interact directly with the TRPV1 C-terminal domain through the K735 residue [31], together with the ARD at the N-terminal end [32]. The ARD domain can also interact with various ligands, including calmodulin, which inhibits TRPV1 channel opening. The calmodulin/ARD interaction is Ca^2+^-dependent and desensitizes TRPV1 following its activation and Ca^2+^ influx [32]. The hormone oxytocin was also shown to activate TRPV1, which consecutively suppresses pain through desensitization of nociceptors and induces hypothermia [33].

Phosphorylation of TRPV1 intracellular domains is achieved by intracellular protein kinases (PKC or PKA) through the activation of G-protein-coupled receptors or tyrosine kinase receptors (TrkA) [7,34]. Phospholipase C hydrolyzes phosphatidylinositol-4,5-biphosphate (PtdIns(4,5)P2 or PIP2) at the plasma membrane to IP_3_ and favors TRPV1 sensitization [34] by the release of PIP_2_ from TRPV1 C-terminal domain [35]. Phospholipase C also promotes the activity of the PKC pathway, which activates TRPV1 by the phosphorylation of two serine residues, one located on the intracellular S2-S3 loop (S502) and the other on the C-terminal domain (S800) [36]. Similarly, the cAMP-dependent PKA phosphorylates Ser116, which causes receptor sensitization [37].

TRPV1 modulation can also occur through the endogenous anandamide agonist, an endovanilloid lipid compound, which is synthesized by several cell types, including neurons. Anandamide was first described as an agonist of the cannabinoid receptor 1 (CB1) and then as a TRPV1 agonist [38]. TRPV1 activation by anandamide is proposed to be involved in major brain functions (see Section 1.3.) regulating motor, visceral, and somato-sensory modalities [39]. The interaction between anandamide and the channel occurs in the CAP vanilloid pocket [40]. When TRPV1 is activated by CAP, prolonged applications with high doses of CAP lead to its Ca^2+^-dependent desensitization and a downstream analgesic effect. This desensitization property is particularly used in the treatment of localized peripheral pain with the use of CAP patches [41]. CAP exerts neurotoxic effects at high concentrations or after long-term use. Similarly to CAP, camphor, a natural compound from the camphor tree (*Cinnamomum camphora*), activates mammalian TRPV1 and desensitizes it robustly, although both agonists are thought to act at different sites [42].

### 1.3. TRPV1 Expression in Mammals

*TRPV1* is expressed in the peripheral nervous system of rodents and humans, including the dorsal root ganglia (DRG) neurons, namely the skin- and viscera-innervating C and Aδ fibers [43]. Among the 17 subtypes of mouse DRG sensory neurons classified by large-scale single-cell RNA sequencing (scRNAseq), *TRPV1* is expressed in five peptidergic DRG neuron subtypes and in two non-peptidergic DRG neuron subtypes [44]. Notably, *TRPV1* is expressed in thermosensitive neurons allowing the transmission of the thermal signal and body thermoregulation, as its activation by CAP causes a modulation of thermal regulation and hypothermia [45]. However, TRPV1 involvement in thermoregulation remains controversial, as *Trpv1*^-/-^ mouse models do not exhibit dysfunction in their thermoregulation [46,47].

The first evidence for functional TRPV1 receptors in human sensory nerves and ganglia was obtained by radioimmunography using [^3^H]-labeled resiniferatoxin (RTX) [48]. With in situ hybridization, TRPV1 was found to be expressed in more than 70% of human sensory neurons, which is significantly higher than in mice (32.4%) [49,50]. In humans, where non-peptidergic neurons do not exist (i.e., all sensory neurons are peptidergic) and nociceptors represent 60 to 70% of all sensory neurons, recent scRNAseq experiments have shown that TRPV1 is highly expressed in:(i)Pruritogen receptor-enriched sensory neurons (involved in itch-sensing);(ii)Putative C low-threshold mechanoreceptors involved in gentle touch (C-LTMRs);(iii)TRPA1-enriched nociceptors (involved in cold, itch pain, and detection of irritants);(iv)Putative silent nociceptors [51,52]—Sensoryomics website https://paincenter.utdallas.edu/sensoryomics/ accessed on 26 January 2025—which correspond to a subset of C-fibers that innervate joints, viscera, and skin, often referred to mechanoinsensitive C-fibers [53]. The latter are unresponsive to noxious mechanical stimuli under normal conditions but can be sensitized after inflammatory stimulation.

*TRPV1* is also expressed in the central nervous system, notably in the hypothalamus, but its precise overall distribution in the brain is still debated [54]. In addition, TRPV1 is expressed in non-neuronal tissues such as pancreas, skeletal muscle fibers, adipocytes, vascular smooth muscle, keratinocytes, and endothelial cells [55,56,57]. It participates in metabolic functions since its activation in these tissues in CAP-treated individuals impacts glucose metabolism, promoting insulin secretion and sensitivity [56,57]. As such, CAP might emerge as a therapeutic target for metabolic syndromes, including symptoms like insulin resistance, obesity, impaired glucose tolerance, and dyslipidemia [58]. Finally, TRPV1 localization is not restricted to the plasma membrane as it was also located in intracellular organelles such as the endoplasmic reticulum or mitochondria [59,60].

### 1.4. Role in Inflammation

Inflammation is consecutive to tissue injury or to the presence of pathogens and can be considered as a “physiological work site” for repairing injuries. Inflammation is concomitant with an increase in temperature and pain development. Several pro-inflammatory agents—such as bradykinin, histamine, and prostaglandins—released by surrounding and immune cells [61], together with an acidification of the local environment, lead to TRPV1 sensitization in rat DRGs [7,62]. Further pro-inflammatory factors, such as lysophosphatidic acid (LPA), are released during neuroinflammation to sensitize TRPV1, resulting in hyperalgesia, which characterizes the inflammatory process characterized in mice [14,63].

When prolonged, inflammation leads to a chronic pain requiring TRPV1 contribution [62,64,65]. For example, TRPV1 participates in the transmission of chronic visceral inflammatory pain due to its expression in DRGs innervating the colon, pancreas, and bladder [47,66]. It also contributes to the pain associated with chronic inflammation during tumorigenesis. In addition, chronic inflammation promotes TRPV1 over-expression, further favoring hyperalgesia and allodynia in humans [67].

### 1.5. TRPV1 Channelopathies

Like most ion channels, *TRP* genes are the target of genetic mutations causing human diseases. Until recently, no channelopathy linked to *TRPV1* variants was described in humans. However, two pathologies associated with pain and thermoregulation alterations have now been identified (Figure 1A and Table 1). Two rare human TRPV1 amino-acid changes (N331K and K710N) were associated with a loss of function in pain syndromes, highlighting TRPV1’s central role as a pain transducer [68,69]. More specifically, the *TRPV1^N331K^* mutation results in a functional knock-out, with insensitivity to CAP and an increased pain threshold to heat (hyposensitivity) and cold (hypersensitivity). Carriers of this mutation also exhibit excessive sweating. Finally, this variant is resistant to an application of RTX or double-knot toxin (DkTx) at concentrations used to activate wt TRPV1. This mutation is located in the ARD, which does not correspond to the CAP binding site (Figure 1A), indicating (i) that this ARD domain is involved, directly or indirectly, in channel activation and pore opening and (ii) that channel insensitivity to agonists involves several parts of the protein, confirming its allosteric pharmacology [68].

In addition, the K710N mutation is located in the intracellular TRP domain (Figure 1A and Table 1). This *TRPV1^K710N^* missense variant exhibits a reduced sensitivity to CAP and CAP-associated acute and inflammatory pain. The knock-in mouse reproducing this human mutation remains sensitive to hot temperatures and does not present any thermoregulation defects with an internal temperature comparable to wt. However, it exhibits reduced sensitivity to neuropathic pain. Given the position of residue 710 in the TRP domain, it appears to be a crucial amino acid for subunit interactions in close proximity to the pore and very probably contributes to the functional coupling between CAP sensitivity and channel opening [69]. These two mutations with total or partial loss of function in pain paradigms—notably in vivo and in vitro sensitivity to CAP—offer interesting prospects for the therapeutic targeting of future analgesics since in both cases no significant alteration in body temperature was observed.

On the other hand, people with a deregulation of their core temperature after physical exercise, known as exertional heat stroke (EHS), may present *TRPV1* variants. In this respect, two mutations have recently been identified in young soldiers whose temperature was equal to or greater than 40 °C during an EHS episode, ultimately with muscular sequelae [70]. The first mutation, G684V, in the sixth transmembrane segment bordering the pore (Figure 1A and Table 1) induces a loss of function of the channel, which is insensitive to CAP in vitro. The second one, R772C downstream of the TRP domain, results in a partial gain of function of the channel when exposed to CAP. For the moment, the sensitivity to acute or inflammatory pain is unknown in carriers of these mutations. These observations are important because they highlight the role of TRPV1 in physiological functions other than pain, particularly in the regulation of internal temperature. This pathophysiological feature associated with TRPV1 variants has also been described in patients with malignant hyperthermia, which manifests with symptoms comparable to heat stroke, such as sudden rise in body temperature, rhabdomyolysis, and loss of consciousness. It is caused by exposure to inhaled volatile anesthetics (VA) or some muscle relaxants. The N394del and T612M variants in TRPV1 were identified in malignant hyperthermia (MH) patients (Figure 1A) exhibiting an increased in vitro sensitivity to the anesthetic halothane [71]. Heterologous expression of these variants in HEK293 cells displayed an insensitivity to CAP and a reduced Ca^2+^ response to isoflurane. However, when expressed in muscle cells, both variants displayed an enhanced sensitivity to isoflurane, confirming the role of TRPV1 in Ca^2+^ homeostasis after stimulation with VA. The different pathological variants of TRPV1 are shown in Figure 1 and Table 1.

**Table 1 toxins-17-00064-t001:** Human TRPV1 subunit variants and associated channelopathies (see also Figure 1A).

Variant	Position in Protein	Human Disease	Clinical Features	Pharmacological Signature	Ref.
N331K	N-terminal 5^th^ ankyrin finger	Pain insensitivity	Loss of function in pain syndroms: heat hyposensitivity and cold hypersensitivity, extensive sweating, no aversion for CAP-containing food	Insensitivity to CAP/RTX/DkTx, acidic pH, heat	[68]
N394del	N-terminal domain	MH	Generalized hyperthermia after exposure to VAs	Insensitivity to CAP, impaired Ca^2+^ homeostasis after VA	[71]
T612M	Pore turret	MH	Generalized hyperthermia after exposure to VAs	Insensitivity to CAP, impaired Ca^2+^ homeostasis after VA	[71]
G684V	S6 transmembrane domain	EHS	Generalized hyperthermia after prolonged and intense activity	Insensitivity to CAP, in vitro muscle contracture after halothane or caffeine	[70]
K710N	TRP domain	Pain insensitivity	Loss of function in pain syndrome (mouse): CAP-induced hyposensitivity, reduced neuropathic pain, no aversion for CAP-containing food	Insensitivity to CAP	[69]
R772C	CaM binding domain	EHS	Generalized hyperthermia after prolonged and intense activity	In vitro muscle contracture after halothane	[70]

## 2. Toxins Acting on TRPV1

Given its role in pain pathways, TRPV1 is now considered a key therapeutic target for the management of acute and inflammatory pains [72]. Antagonistic strategies focusing on this receptor have been developed for years but displayed the drawback of side effects, mainly hyperthermia [73,74]. On the other hand, most agonists used to initiate desensitization of the receptor lead to hypothermia [75]. Therefore, a better understanding of the pharmacology of this receptor is essential.

Exogenous agents acting on TRPV1 include natural compounds produced by plants and animals. In animal venoms, toxins active on TRPV1 are supposed to play a crucial role in predation/defense strategies. In the case of activators, their function consists in inflicting pain on prey or predator or in warding off a potential danger [76]. Conversely, other toxins are known to have specific TRPV1-blocking activity, but their ecological and evolutionary roles remain to be clarified. We review TRPV1-binding toxins, acting either as agonists or antagonists, and their known or proposed interaction sites on the channel (Figure 2). The diversity of allosteric sites illustrates on one hand its versatile and polymodal function and on the other hand the diversity and specificity of natural toxins to bind singular domains of this receptor. These toxins are invaluable pharmacological tools which provide innovative routes to design novel treatments for acute and inflammatory pains [77].

### 2.1. Agonists

The discovery of TRPV1 agonists in venoms and plants suggests that pain is an important sensory system in defense strategies against intruders and predators. Several of the toxins presented below are highly specific for TRPV1, sometimes with a high affinity, providing natural tools to characterize the allosteric pharmacology of TRPV1.

#### 2.1.1. RTX

Resiniferatoxin (Figure 2) is a diterpene produced in the latex of different cactus species belonging to the genus *Euphorbia* (*E. resinifera, E. poissonii,* and *E. unispina*, Figure 3). As a TRPV1 agonist, it shares a common domain with CAP, the 4-hydroxyl-3-methoxyphenyl moiety [79]. Conversely, an iodinated RTX derivative named I-RTX is a TRPV1 antagonist [80]. RTX induces similar effects to CAP, such as hypothermia, neurogenic inflammation, and pain [81], although with a much more potent effect (EC_50_ = 39 nM on currents in TRPV1-expressing oocytes—Table 2) [9], ultimately leading to high Ca^2+^ entry in cells causing sensory neuron death [82]. Intrathecal injections of RTX were performed in Sprague Dawley rats and revealed a selective TRPV1 depletion in neuronal afferents. A similar decrease in TRPV1 expression was also observed from a behavioral point of view in rats displaying an increase in the thermal and mechanical pain thresholds related to inflammation [83].

The interaction of RTX with the human and rodent TRPV1 proteins was initially characterized by binding assays to identify the key residues involved (Figure 3). The RTX binding domain is located intracellularly (Figure 4). The cryo-EM structure of TRPV1 bound to RTX enabled the precise identification of each binding site [25]. It involves amino acids of the S2-S3 linker (Y511-S512) and of the S4 transmembrane domain (M547, T550). Interestingly, the rat receptor is much more sensitive to RTX than the human ortholog [84] due to the M547 (Figure 3), located on the S4 transmembrane segment [85], while human TRPV1 (hTRPV1) displays an L547. This difference between species led to the conclusion that the vanilloid pocket responsible for TRPV1 activation by agonists includes this amino-acid position and that its mutation to leucine drastically decreases RTX affinity for TRPV1 [85]. RTX-induced analgesic effect is in part due to its high affinity for TRPV1 and the modulation of its expression [83,84]. The therapeutic use of RTX is currently envisaged in two ways: desensitization of TRPV1, which is a reversible process, and the death of nociceptors, which is obviously irreversible [86]. It reached Phase III in clinical trials to evaluate its efficacy for the management of knee osteoarthritis-induced pain (https://clinicaltrials.gov/search?term=resiniferatoxin, accessed on 3 January 2025).

#### 2.1.2. Vanillotoxins (VaTx)

Spiders are active venomous animals that inoculate their venom offensively for predation or defensively to escape a predator or an intruder. Spider venoms are important reservoirs of toxins active on a wide range of targets. Three vanillotoxins (VaTx1-3) from the venom of a Caribbean spider, *Psalmopoeus cambridgei* (Figure 3), target the outer part of the TRPV1 pore domain (extracellular loops between S5-S6), causing its activation and in vivo painful responses [87]. Intraplantar injection of VaTx3, the most potent, produces painful symptoms in mice, comparable to what is observed with crude venom injection. VaTx1-3 selectively activate TRPV1 without effect on TRPA1 or TRPM8. These peptidyl toxins exhibit an inhibitor cystine knot (ICK) motif (Figure 2), i.e., a structural feature widely described in spider, conus, and in some scorpion venoms, as a modulator of cation channels, similar to hanatoxins or κ-conotoxin PVIIA for potassium channels (Kv). The toxin ICK motif is rigid and compact and composed of 26 to 48 residues and three disulfide bonds [87,88,89]. High concentrations of VaTx1 also inhibit Kv2 channels, a property already described in homologous spider venom peptides sharing target promiscuity, which explains their contribution to neuronal hyperexcitability [90]. VaTx3 shares a structural similarity with hanatoxins (HaTx) from the venom of another spider (*Grammostola rosea*, formerly *G. spatulata*) [91,92]. It is also more effective than VaTx1 and VaTx2 in increasing intracellular Ca^2+^ in TRPV1-expressing HEK-293 cells, as demonstrated by EC_50_ assessment (VaTx3: 0.45 μM, VaTx2: 1.35 μM, and VaTx1: 9.9 μM—Table 2). VaTx1, the least active on TRPV1, blocks Kv2.1 channels in a voltage dependent manner. Interestingly, in the same venom, the analgesic peptide PcTx1 blocks other pain-related channels, namely ASICs [93]. In situ injection of VaTx3 causes a pain-related reaction in wild-type mice, which is absent in *Trpv1*^-/-^ mice. As with CAP injection, this reaction generates a local inflammation leading to edema [87].

#### 2.1.3. Double-Knot Toxin (DkTx)

The Chinese bird spider *Cyriopagopus schmidti* (formerly *Ornithoctonus huwena,*
Figure 3) is a tarantula producing a 79-amino-acid toxin which activates TRPV1. It is a “double-knot” toxin (DkTx) because of its bivalent structure based on two lobes (K1 and K2), each with an ICK motif (Figure 2). The DkTx/TRPV1 complex is very stable and almost irreversible, with a low dissociation rate caused by the toxin’s bivalence. Compared to CAP, the persistence of a TRPV1 current can be observed following the action of DkTx [92]. The toxin is strongly selective towards TRPV1 as it does not interact with Kv, Nav, and Cav channels nor with other members of the TRP channel family.

DkTx bivalence is characterized by the toxin’s ability to bind two receptor sites, locking TRPV1 in an open conformation: each toxin lobe K1/K2 interacts with the same site on two adjacent TRPV1 subunits [92,94,95]. Both sites are localized at extracellular positions critical for the channel function, determined by cryo-EM of TRPV1 bound to DkTx [25]: (i) the transmembrane segment S6 responsible for toxin binding and (ii) the pore-forming S5–S6 loop (including residues I599, F649, A657, F659) (Figure 3). This TRPV1 domain is crucial to regulate the opening and permeability of the channel [92,95,96] (Figure 4). Altogether, these data shed light on TRPV1’s biophysical properties and its activation by allosteric mechanisms. Moreover, one of the pathological variants of hTRPV1 (N331K, located in the N-terminus) is insensitive to DkTx [68], adding to the allosteric character of this receptor (Table 1).

rTRPV1 channels have been functionally expressed in worms’ polymodal nociceptive neurons to confer a specific, robust, and dose-dependent avoidance behavior to CAP. However, DkTx which also binds to rTRPV1 does not elicit aversive behavior in *C. elegans* even at high concentrations [96]. In fact, DkTx activates rTRPV1 but its effects on nociception are still unknown in rodents and further study will be required. The presence of two analgesic peptides (μ-TRTX-Hh2a and ω-TRTX-Cs16) active on the Nav and Cav channels has been described in the same spider venom, which complexifies the interpretation of their role with respect to pain [97].

#### 2.1.4. BmP01

Produced by the scorpion *Mesobuthus martensii*, in far East Asia, the neurotoxin BmP01 (Figure 2) also contains an ICK domain, like DkTx/VaTx, and activates TRPV1 current with an EC_50_ of 131.8 and 40.4 μM in mTRPV-1 and hTRPV1-expressing HEK293T cells, respectively (Table 2). This peptide includes 29 amino acids cross-linked by three disulfide bonds, generating an α-helix and two β-sheets [98,99], which confer a high specificity for TRPV1. Functionally, scorpion venom has a natural acidic pH (6.5), synergizing by a bimodal process the activation of nociceptors through TRPV1 protonation [99], and can increase the toxin affinity for TRPV1, which explains its low affinity at physiological pH. Although the toxin was shown to inhibit Kv channels, pH acidification decreases this inhibition. BmP01 injection in mice evokes acute pain responses reflecting its specific effect on peripheral TRPV1, as this pain sensation is absent in *Trpv1*^-/-^ mice. It is also interesting to note that *Mesobuthus martensii* venom contains several analgesic toxins able to block the voltage-gated Na^+^ channels (Nav1.7; Nav1.8) implicated in pain [97]. To activate TRPV1, BmP01 interacts with the channel polar residues in the loop prior to the S6 transmembrane domain (E649: proton activation site, T651, E652) (Figure 3 and Figure 4) to open the pore and promote the consecutive envenomation pain [99]. The K23 of BmP01 directly interacts with E649 of TRPV1. The synergy between the toxin and the pH environment confirmed the polymodal nature of the TRPV1 channel. The binding site is thus extracellular, close to the pore domain.

**Figure 3 toxins-17-00064-f003:**
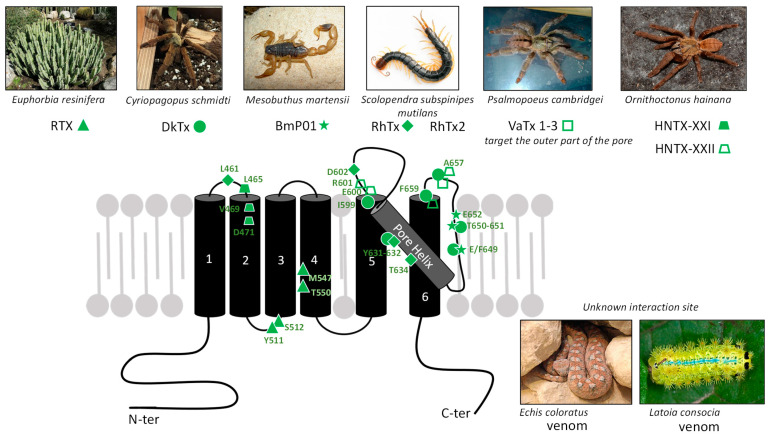
Agonist toxins and their binding sites on a mammalian TRPV1 subunit. Each toxin is associated with its venomous or poisonous producer. When they are known or proposed, each amino-acid residue of TRPV1 involved in toxin bindings site is displayed with its protein position (green numbers). For more details, see [25] for RTX (▲), [87] for VaTx (
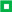
), [92] for DkTx (●), [99] for BmP01 (
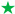
), [100,101] for RhTx (♦), and [102] for HNTX-XXI (
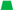
) and HNTX-XXII (
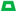
). *Latoia consocia* is from ©Daniel Ruyle. All other images are from Wikimedia commons.

#### 2.1.5. RhTx Toxin

RhTx is a 27-amino-acid peptide (Figure 2) from the venom of *Scolopendra subspinipes mutilans*, also called Chinese red-headed scolopendra (Figure 3). NMR spectrometry revealed the structure of this compact peptide, including two disulfide bonds, a flexible N-terminal, and a polarized C-terminal domains [100]. Behavioral studies in mice exposed to this toxin showed a pain induction which is similar to the one produced by CAP and abolished in *Trpv1*^-/-^ mice. RhTx binding to TRPV1 is rapid, while the desensitization is very slow, thus favoring persistence of the TRPV1-mediated current [100]. In vitro, RhTx targets TRPV1 with an estimated EC_50_ of 470 nM (current in rTRPV1-HEK293T cells) to 521 nM (current record in DRG neurons—Table 2). At a lower concentration of 100 nM, RhTx decreased the temperature activation threshold of TRPV1, from 43 to 37 °C, corresponding to the mammalian body temperatures. On the other hand, cold temperatures inhibit this RhTx-induced TRPV1 activation, in contrast to CAP [100]. It makes RhTx a facilitator of heat-dependent activation of TRPV1, confirming the high polymodality of the channel. Consequently, RhTx lowers the thermal threshold (of ~6 °C), which results in the normal body temperature of a mammal being higher than the toxin-modified temperature threshold of the channel. As expected, intraplantar injection of RhTx in mouse induced—like CAP—a decrease in core temperature.

RhTx residues responsible for the toxin–channel interaction are charged residues located at the C-terminal domain, forming electrostatic and hydrophobic interactions. Their mutation leads to either a higher (R15A) or a lower (D20A, K21A, Q22A, E27A) affinity for TRPV1. These amino acids interact on the outer side of the channel, including the pore-forming helix (Y632 and T634), the turret preceding the pore helix (D602), and the extracellular loop located between S1 and S2 (L461) [100] (Figure 3).

RhTx2, another toxin from the venom of *Scolopendra subspinipes mutilans* exhibits the same structure as RhTx, with four additional amino acids at the N-terminal of the peptide, arranged into a positively charged helical conformation (K3, R19, and K30). Disulfide bonds are formed between C9 and C20 and C14 and C27 [101]. However, RhTx2 is less efficient than RhTx to activate TRPV1, as shown by EC_50_ values in TRPV1-expressing HEK293T cells (38.3 μM for RhTx2—Table 2). This can be explained by differences in affinity, resulting in a faster channel desensitization with RhTx2 due to a lower probability of the pore opening and the reduction in its conductance [101]. Consequently, RhTx2 is an interesting pharmacological tool to address the mechanisms of TRPV1 desensitization.

**Figure 4 toxins-17-00064-f004:**
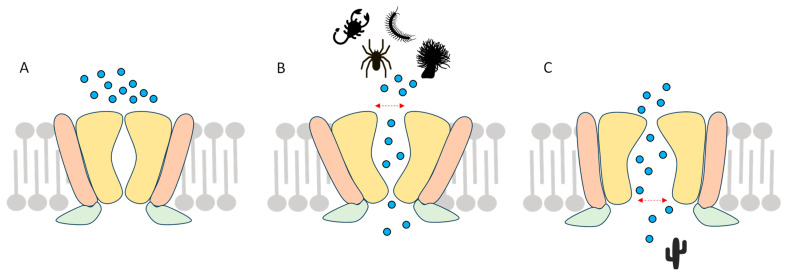
Schematic representation of TRPV1 opening by various agonist toxins. (**A**) Closed state with cations in the extracellular domain (blue dots). The pore domain is depicted in yellow, S1-S4 transmembrane domains in orange, and the ARD in green. For simplicity, only two subunits are shown. (**B**) Animal toxins widen the upper pore (red arrow), allowing cations to flow through the channel. (**C**) Vegetal toxins expand the lower gate through intracellular binding (red arrow).

#### 2.1.6. Hainantoxins (HNTX-XXI and HNTX-XXII)

Recently, two new peptidyl toxins from the venom of the Chinese spider *Ornithoctonus hainana* (Figure 3) were identified [102]. They were named HNTX-XXI (64 amino acids, four disulfide bridges) and HNTX-XXII (77 amino acids, six disulfide bridges). These two peptides appear to be TRPV1-specific agonists: HNTX-XXI and HNTX-XXII induce a CAP-sensitive current with an EC_50_ of 3.6 and 0.86 μM, respectively (in TRPV1-expressing HEK293 cells), and do not act on most voltage-sensitive channels, notably Nav, Kv, and Cav. Interestingly, HNTX-XXII shares 86% sequence homology with DkTx and both toxins are organized into two ICK moieties (K1/K2). Site-directed mutagenesis experiments show that HNTX-XXII and DkTx share a common binding site on TRPV1 since several amino acids appear to be part of the pore region on the segment between S5 and S6 (Figure 3). These data suggest that the mode of action of HNTX-XXII and DkTx are identical: they widen the upper part of the pore by targeting the outer pore domain. As for HNTX-XXI, its binding site—determined by site-directed mutagenesis—is located in the S1-S2 loop and on the upper part of S2. Notably, the presence within the same venom of two peptidyl toxins with different structures and targeting the same receptor at different sites is original.

#### 2.1.7. Viperidae Venom

The venom of the Middle Eastern snake *Echis coloratus*, also known as the Palestine saw-scaled viper (Figure 3), contains proteins exhibiting an agonist effect on TRPV1. Some fractions extracted from the venom induce cellular responses related to TRPV1 activation, such as a Ca^2+^ influx in TRPV1-expressing HEK293 cells, which is abolished in *Trpv1*^-/-^ cells. Compared to CAP, one of these fractions caused a comparable, albeit lower, amplitude, outwardly rectifying TRPV1-mediated currents. Such an effect seems to be specific to TRPV1 because it was not observed when this fraction was tested on TRPA1-, TRPM8-, or TRPV2-expressing HEK293 cells [103]. The binding site of these venom peptidyl fractions on TRPV1 might be different from RTX and the spider toxin interaction sites, as mutations affecting residues crucial for RTX and VaTx/DkTx binding do not modify the amplitude of the TRPV1 current generated by the active fraction of this venom. Importantly, the nerve growth factor (NGF) present in the *Echis coloratus* venom acts in synergy with the toxin, directly activating TRPV1 and producing a robust pain during envenomation.

#### 2.1.8. Caterpillar Venom

The venom of the caterpillar *Latoia consocia* has recently been studied for its possible action on TRPV1 (Figure 3). In caterpillars, the venomous function is often associated with urticating setae [104]. A behavioral study using intraplantar venom injection in wild-type mice showed acute pain induction related to a direct activation of TRPV1, since in vivo experiments disclosed the absence of pain in *Trpv1*^-/-^ mice and decreased pain intensity in wild-type mice following the pre-treatment with CPZ. In vitro, a significant intracellular Ca^2+^ increase was observed in DRG neurons exposed to the venom (100 μg/mL). The molecule responsible for this TRPV1 activation remains to be identified [105].

Other venoms containing toxins have been proposed to interact with TRPV1 to elicit pain, notably the venom of the armed spider *Phoneutria nigriventer* (Figure 5). The pain induced in mice is the result of different toxins targeting several receptors, including the TRPV1, ASIC, and Nav channels, all known to mediate pain in sensory neurons. The implication of TRPV1 was proposed after the inhibitory effect of the antagonist SB366791 on venom-elicited in vivo pain [106].

**Table 2 toxins-17-00064-t002:** Animal and vegetal toxins modulating TRPV1 (see text for details). Agonists are framed in green, antagonists in red.

Molecule	Effect on TRPV1	EC_50_/IC_50_	Mode of Action	Site of Action	Identification or Prediction of the Site of Action	Ref.
CAP	Agonist	712 nM	Stabilizes the open state	S3, S4	Cryoelectromicroscopy	[9,25]
RTX		39 nM	Stabilizes the open state through expansion of the lower gate	S3, S4	Cryoelectromicroscopy	[9,25]
DkTx		230 nM	Widens the upper pore	Outer pore domain	Cryoelectromicroscopy, site-directed mutagenesis	[25,92,94]
VaTx1 VaTx2 VaTx3		9.9 µM 1.35 µM 0.45 µM	Widens the upper pore	Outer pore domain	Site-directed mutagenesis	[87,92]
BmP01		131.8 µM 40.4 µM	Widens the upper pore in acidic pH	Outer pore domain	Site-directed mutagenesis	[98,99]
RhTx RhTx2		470–521.5 nM 38.3 µM	Widens the upper pore in a T°-dependent manner	Outer pore domain + S1/S2 loop	Site-directed mutagenesis	[100,101]
HNTX-XXI HNTX-XXII		3.6 µM 0.86 μM	Widens the upper pore	Outer pore domain	Site-directed mutagenesis	[102]
AG489	Antagonist	300 nM	Pore blocking	Pore-forming extracellular loop	Site-directed mutagenesis	[107]
PnTx3-5		30 nM	nd	nd	nd	[108]
APHC1 APHC2 APHC3		60 nM 23 NM 18 nM	Stabilize an intermediate state at receptor activation	Pore helix and outer pore domain; S1/S2 and S3/S4 external loops	Molecular modeling	[109]
HCRG21		6.9 µM	nd	Outer pore domain	Molecular modelling	[110]
Tst2		100 nM	nd	nd	nd	[111]

### 2.2. Antagonists

Molecules that antagonize TRPV1 were initially called “toxins” because they have been discovered in animal venoms, but their toxic effect has usually not been characterized and the benefits of their presence in venoms remains unknown. If specific to TRPV1, they could be used as pharmacological bases for the development of analgesic agents.

#### 2.2.1. AG489

Spider venom toxins have been shown to antagonize TRPV1 as well. The first toxins discovered are two acylpolyamine molecules, named AG489 (Figure 2) and AG505, extracted from the venom of *Agelenopsis aperta*, categorized as a funnel-web spider (Figure 5) [107]. These toxins are full TRPV1 antagonists since they completely inhibit CAP-induced currents in TRPV1-expressing oocytes (IC_50_ around 300 nM for AG489—Table 2) using a pore-blocking mechanism. They share the same structure (AG505 holds an extra hydroxide on its first cycle). Despite their robust antagonist effect, they are not specific to TRPV1 as AG489 and AG505 are also known to be antagonists of NMDA-type glutamate receptors [112].

The interaction between AG489 and TRPV1 was investigated using channel-directed mutagenesis and revealed the role of the S5-S6 linker region in toxin binding. In this respect, the modification of N628, E636, D646, or E651 increases TRPV1 sensitivity, whereas the modification of Y627 or C634 displayed the opposite effects [107]. All these residues are located in the pore extracellular region between S5 and S6, supporting that AG489 inhibition of TRPV1 can be explained by decreased ion conduction, pointing to an interaction between the toxin and the pore-forming extracellular loop (Figure 4).

#### 2.2.2. PnTx3-5

More recently, another TRPV1 antagonist toxin was identified in the venom of the armed spider *Phoneutria nigriventer* (Figure 5). Although this venom mediates pain behavior after intraplantar injection in mice [106], it also contains an antinociceptive peptide, named PnTx3-5 (Figure 2), which is used in vivo to alleviate neuropathic and cancer-related pains [113]. Its analgesic potency varies depending on the pain model used, and a maximum effect was reached with PnTx3-5 intrathecal injection of 30 fmol/site in mouse post-operative models. Moreover, a local injection of PnTx3-5 (100 fmol, intradermal) prevents the nociceptive behavior induced by CAP injection into rat left vibrissae in an orofacial test [108]. No mechanical hyperalgesia was observed following the administration of this toxin in the post-operative pain paradigm, even in morphine-tolerant mice [113]. PnTx3-5 is allegedly an alternative to opioids without notable side effects, such as an alteration in locomotor activity. The analgesic effect observed is mediated by blocking the pain signal in sensory neurons. In rat trigeminal ganglia, PnTx3-5 dose-dependently inhibited CAP-induced glutamate release with maximal inhibition at a 100 nM concentration (IC_50_ 47 nM). PnTx3-5 exhibited a selective effect on TRPV1 in vitro as it inhibited both the Ca^2+^ rise and the current elicited by CAP in TRPV1-expressing HEK293 cells with an IC_50_ of ~30 nM [108] (Table 2). To date, the precise toxin binding site on the channel remains to be determined. Several peptides in *Phoneutria nigriventer* venom (PnTx3-3, Phα1β, PnTx4(5-5)) are proposed to act in the same analgesic way by blocking other various channels involved in pain, such as TRPA1, Cav2.2, and NMDA-R [97].

#### 2.2.3. APHC1-3 Polypeptides

The sea anemone *Heteractis crispa* (Figure 5) produces three venom-containing polypeptides, APHC1, APHC2 and APHC3, targeting TRPV1. Their primary structure shows strong homologies with the Kunitz-type trypsin inhibitors, with a distinctive motif present in most serine protease and ion-channel inhibitors [114,115] (Figure 2). APHC1-3 are composed of 56 amino acids and exhibit a high degree of sequence homology. Even though they act as partial antagonists of TRPV1 with a maximal inhibition of 32% in TRPV1-expressing oocytes stimulated with CAP [114], APHC1 and APHC3 display significant antinociceptive and analgesic properties. Intravenous or intramuscular APHC1 injection decreases the acute pain induced by an intraplantar CAP injection or by tail immersion in hot water, respectively. Intravenous administration of these toxins also blocked formalin-induced behavior, reversed CFA-induced hyperalgesia, and produced hypothermia. APHC3 displays a slight sequence difference in four residues with APHC1. It causes a reversible inhibition of the CAP-induced current in TRPV1-expressing HEK293 cells. Its binding to TRPV1 significantly blocks a current induced by acidic pH (5.5), with a maximal inhibition of 77% at 240 nM [116]. To date, APHC1-3 are considered as partial TRPV1 antagonists, as revealed by various in vitro studies [109,114]. This specific inhibitory effect was used in vivo to induce analgesia but without inducing hyperthermia like most TRPV1 antagonists do [116]. Interestingly, in rTRPV1-expressing CHO cells, these toxins were shown to potentiate CAP-induced TRPV1 currents at low concentrations (< 1μM) and to antagonize the same effect with high CAP concentrations (>3 μM) with IC_50_ of 60, 23, and 18 nM for APHC1-3, respectively (Table 2—IC_50_ was 54 nM for APHC1 in hTRPV1-expressing *Xenopus* oocytes, as published by [109]). These opposing effects highlight the allosteric nature of TRPV1 and its ability to bind different ligands at different sites, resulting in either potentiation or inhibition.

The interactions between APHC toxins and TRPV1 were proposed after molecular modeling and suggested a putative binding site at the TRPV1 outer loops [109]. The APHC C-terminal helix interacts with the pore helix and with the two extracellular loops of TRPV1. Importantly, binding sites are dependent on TRPV1 conformational rearrangements and vice versa, and toxin binding can affect the balance between the different states of the TRPV1 channel (closed, intermediate, or open) [109].

#### 2.2.4. HCRG21

HCRG21 is a 56-amino-acid peptide also produced by the same sea anemone species, *Heteractis crispa* (Figure 5). It shares common structural properties with APHC Kunitz-type peptides and a high degree of sequence homology with APHC1-3 (89 to 93%). Like other Kv channel-inhibitor toxins, HCRG21 possesses positively charged residues throughout its sequence, but it does not inhibit K^+^ current in Kv-expressing oocytes [110]. HCRG21 exhibits a potent TRPV1 antagonist effect with a maximum 95% inhibition of the CAP-induced current at 100 μM (IC_50_ 6.9 μM—Table 2), as evaluated in *Xenopus* oocytes. In vivo, HCRG21 exhibits analgesic effects against pain in mice. It reduces local inflammation and mechanical and thermal hyperalgesia, all induced by carrageenan intraplantar administration together with a decrease in the inflammatory cytokine TNF-α in the blood [117].

Molecular modeling further suggested different possible binding sites of HCRG21 on TRPV1. Indeed, HCRG21 targets residues of the channel pore outer vestibule, close to the DkTx binding site (Figure 5 and Figure 6) close to the F649-E651 domain [110]. Considering the TRPV1 amino acids involved, HCRG21 could be a pore blocker, unlike DkTx, which activates TRPV1 and locks it in an open conformation. This interaction is based on the binding of R48 from HCRG21 to E648 and D646 of one TRPV1 subunit, as well as E636 of the adjacent TRPV1 subunit, through hydrogen bonds and multiple attractive electrostatic interactions. Furthermore, according to molecular modeling, R48 from HCRG21 can form a hydrogen bond with M644 from another TRPV1 subunit. Possible other HCRG21 residues like R1, R18, K28, R51, and R55 may be crucial for TRPV1 binding. These interactions could induce a conformational change of the channel gating and consequently the blockade of the cation influx. As they are based on molecular modeling, they need to be confirmed by structural or functional studies.

#### 2.2.5. Tst2

Another sea anemone peptide has recently been identified: it is produced by the tropical species *Telmatactis stephensoni*, found in Australia. This 38-amino-acid peptide, named Tst2, has an ICK-type structure and three disulfide bonds [111] (Figure 2). It is likely that this peptide is synthesized and excreted in the venom of this sea anemone, although this remains to be determined. The peptide sequence of Tst2 shows 36–42% sequence homology with spider venom toxins, notably Pn1a from *Phoneutria nigriventer*. These toxins are modulators of Nav channels [118].

The electrophysiological study of the Tst2 peptide (100 nM) showed in CHO cells that it did not act on the currents dissipated by the human Nav1.3-5 and Nav1.7 channels, nor on the Kv2.1 channel, nor on voltage-gated proton channel Hv1. On the other hand, at this concentration, it blocked more than 50% of the TRPV1 current activated by CAP (1 μM) [111]. This result is rather counterintuitive, given that envenomations involving *Telmatactis stephensoni* are quite painful. At present, the receptor site of the Tst2 peptide at the TRPV1 channel is not known.

## 3. Conclusions

Over the course of evolution, animals and plants produce numerous compounds displaying a high specificity for certain receptors. In the last decade, many of them were shown to target TRPV1. Such exogenous ligands provide an interesting toolbox to understand the channel pharmacology as they interact with several functional regions, including the pore helix, intracellular, extracellular, and transmembrane domains (Figure 6), with distinct activation and inhibition mechanisms. Indeed, toxin binding sites on TRPV1 influence its function in different ways depending on the targeted domain. Thus, agonists highlight the crucial interaction sites for positive TRPV1 modulation, such as the intracellular vanilloid pockets or the different extracellular sites for ICK peptides (Figure 4). These agonistic toxins discriminate the channel in different ligand-dependent opening states. Moreover, as reviewed here, animal toxin binding sites affect the pore helix, highlighting its importance in TRPV1 opening. However, most toxins binding this domain do not display the same effects on TRPV1 function. For example, several toxins (AG489, APHC, HCRG21) revealed their ability to inhibit Ca^2+^ influx, while others (RhTx) favored the persistence of the TRPV1-mediated current, all of them binding the pore region. The identification of key residues involved in the binding of different toxins, coupled with the characterization of the conformational and functional modifications, are important to understand TRPV1 structure/function correlations. Knowledge of the interaction and activation mechanisms is an essential concern for further understanding TRPV1 modulation, either positively or negatively (Figure 6). Thus, these toxins are excellent pharmacological tools to map the channel at the residue level and characterize its biophysical properties and allosteric functions.

In addition, the studies of these toxins highlight their therapeutic potential in targeting TRPV1 to manage pain, which in our societies is essentially based on pharmacologically active molecules acting on nociceptive mechanisms. Nowadays, opioids are the most widely used drugs for treating intense and chronic pain. However, they cause different side effects, which compromise their long-term use and justify the development of new treatments. In 2004, the first animal neurotoxin, ω-contoxin MVIIA (Prialt^®^) from *Conus magus*, a specific inhibitor of N-type Ca^2+^ channels (Cav2.2), obtained FDA approval for the treatment of chronic refractory pain. But, its use is complex because it requires intrathecal administration and careful dose titration to avoid neurological intoxication due to its narrow therapeutic window [119]. The use of TRPV1 agonists is a promising avenue for pain management since CAP and RTX are currently molecules whose analgesic activity is based on pharmacological effects linked to receptor desensitization or cell death.

The antagonistic strategy targeting TRPV1 is also considered against inflammatory or chronic pain, where the drug decreases hyperalgesia and increases the threshold for detecting noxious heat. As TRPV1 inhibitors, they exhibit a therapeutic potential in inflammatory pain, although synthetic TRPV1 antagonists induce hyperthermia in human [120]. In this respect, recent data have shown that TRPV1 from sensory neurons transduces antagonist-induced hyperthermia and agonist-induced hypothermia [121]. This blackspot could be circumvented by the observation that APHC toxins provide an analgesic effect but no hyperthermia. This is illustrated by the in vivo analgesic and anti-inflammatory APHC3 properties, which reduce mechanical and thermal hyperalgesia after subcutaneous administration in rats [122]. Thus, TRPV1 antagonists could further be used to relieve chronic pains experienced in neuropathy and cancer, and, as an example, PnTx3-5 treatment may overcome the use of morphine due to its absence of in vivo side effects, such as locomotor disorder or tolerance. On the other hand, one could take advantage of the side-effects of TRPV1 agonists—namely hypothermia—to effectively manage pathologies inducing hyperthermia-like heat stroke, for which no approved drug currently exists. This therapeutic avenue deserves to be pursued in the future.

## Figures and Tables

**Figure 1 toxins-17-00064-f001:**
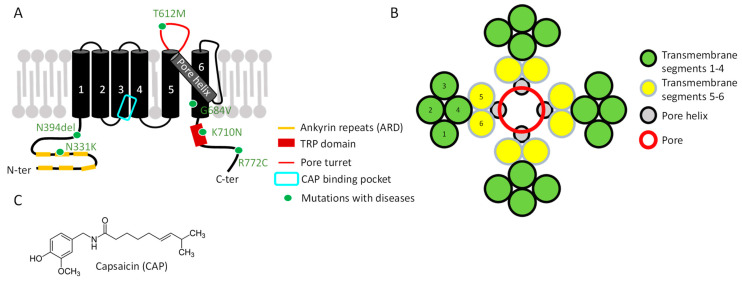
The mammalian TRPV1 channel. (**A**) Schematic representation of one TRPV1 subunit with the 6 transmembrane segments (S1-S6), a pore turret (red), a pore helix between S5 and S6, an intracellular N-terminal with an ARD, and an intracellular C-terminal with the highly conserved TRP domain. (**B**) Schematic representation of TRPV1 homotetramer. The S5-S6 pore helix of each subunit is linked to the adjacent subunit to delineate the channel’s pore. (**C**) Structure of CAP, the main agonist of TRPV1. See also [12].

**Figure 2 toxins-17-00064-f002:**
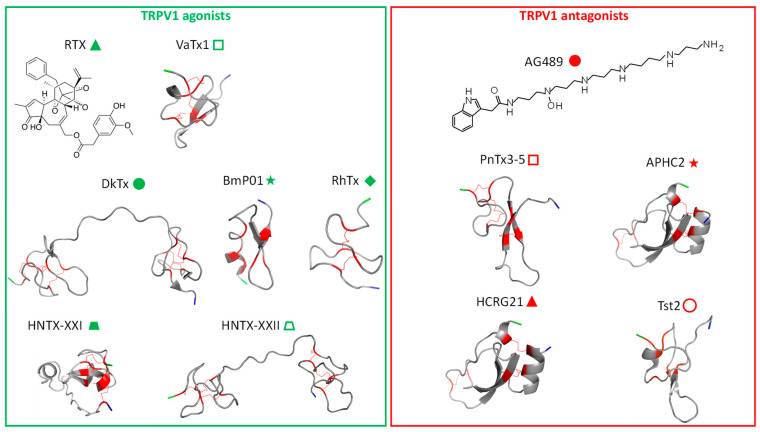
Biochemical and 2D structures of some TRPV1-activating (green symbols) and -inhibiting (red symbols) toxins. HNTX-XXI and HNTX-XXII structures were obtained by homology modeling on the template Hainantoxin-XVIII-7 from *Haplopelma hainanum* (sequence identity 98.44%; GMQE 0.69) and Tau-Liphistoxin Lth1a_1 from *Liphistius thaleban* (sequence identity 92.11%; GMQE 0.87), respectively, using Swiss-Model server [78]. PDB ID, AlphaFold ID, and Pubchem structures were used in PyMOL 3.0 software or ACD/ChemSketch freeware 2022.1.0 to design all other molecules. N-term-, C-term-, and disulfide-involved amino acids are labeled in green, blue, and red, respectively. RTX (Wikimedia commons), DkTx (PDB 6CUC), BmP01 (PDB 1WM7), RhTx (PDB 2MWA), VaTx1: Vanillotoxin-1 (AF-P0C244-F1), AG489, PnTX3-5 (AF-P81791-F1), APHC2 (AF-COHJK4-F1), HCRG21 (AF-P0DL86-F1), Tst2 (PDB 8SEM).

**Figure 5 toxins-17-00064-f005:**
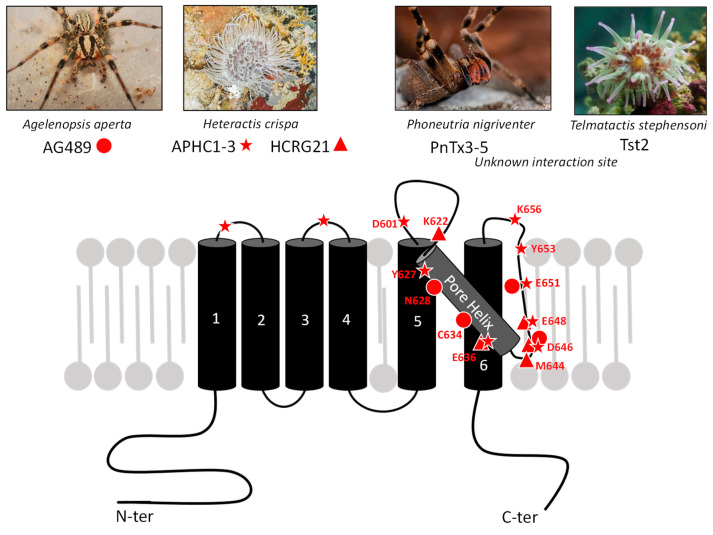
Antagonist toxins and their binding sites on mammalian TRPV1 subunit. Each toxin is associated with its venomous producer. When they are known or suspected, each TRPV1 residue involved in toxin binding is displayed with their respective position in the protein sequence (red numbers). For more details, see [107] for AG489 (●), [108] for PnTx, [109] for APHC (
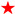
), and [110] for HCRG21 (▲). All images are from Wikimedia commons.

**Figure 6 toxins-17-00064-f006:**
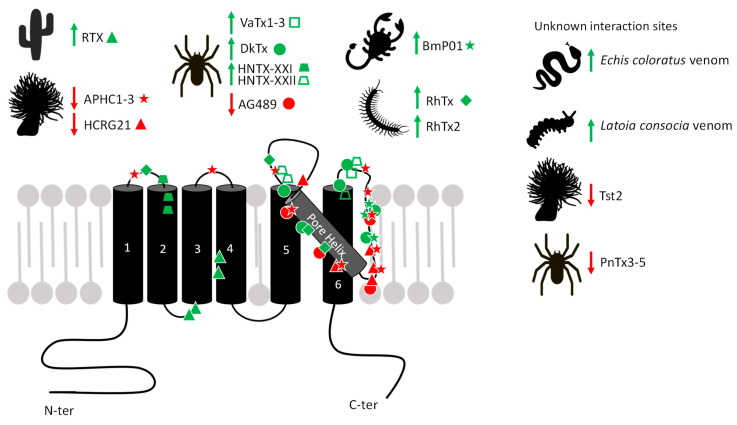
Schematic representation of toxins binding to mammalian TRPV1 subunit, including agonists (green arrows) and antagonists (red arrows) with their ascertained or hypothesized interacting residues. See text for details.

## Data Availability

No new data were created or analyzed in this study. Data sharing is not applicable to this article.

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
