# Peer review of "Neurotoxins Acting on TRPV1—Building a Molecular Template for the Study of Pain and Thermal Dysfunctions"

_toxins, 2025, doi:10.3390/toxins17020064_

Round 1

Reviewer 1 Report

Comments and Suggestions for Authors

The overall composition of the manuscript is good. The paper is scientifically and methodologically accurate. This manuscript will interest many readers. However, I have comments on it. More detailed comments are given below.

General comments

The main problem with this revision is that the figures are not placed according to the order in which they appear in the main document. Reading this document in its current format is complicated because the reader must move several pages forward or backward in the main document. It is suggested that figures be added according to the order of the document. For example, it is suggested that Figure 2 be separated because it presents agonists and antagonists (page 7), while the information regarding the antagonist’s section is located on page 13.

Specific comments

Figure 1. In the Figure legend, add the appropriate reference.

Lines 80 and 82. Please separate the value of pH. i.e., pH5 to pH 5.0

Line 153. Please change to italic TRPV1.

Table 1. It is suggested that authors make more use of the information in their table. It is only mentioned once in the main document.

Figure 2. Authors should explain why different green and red symbols are presented.  Besides, it is suggested that the figure be separated into agonists and antagonists. Furthermore, it is suggested that the agonists be placed in the order of their appearance in the main document. The way they are presented is confusing, and they do not have an order of appearance in the figure. In addition, the use of panels would be a better way of presentation.

2.1.3. Double-knot toxin (DkTx). For a better understanding of the readers, it is suggested that the information corresponding to figure 3 be indicated first and then the information corresponding to figure 4 be added. The way this paragraph is presented is confusing because Figure 4 is indicated first and then Figure 3.

Lines 353-355. Please verify the following sentence: "Both sites are localized at extracellular positions (Figure 4) critical for the channel function, determined by cryo-EM of TRPV1 bound to DkTx [25]: " This information is not related to Figure 4. Please verify. Besides, it is suggested that this information be indicated in Figure 2.

Figure 3. Please include the symbols with the corresponding names in the figure legends.

Lines 560-569. It is suggested that the authors introduce a scheme regarding this paragraph

Author Response

Figure 1. In the Figure legend, add the appropriate reference.

- Done

Lines 80 and 82. Please separate the value of pH. i.e., pH5 to pH 5.0

- Done

Line 153. Please change to italic TRPV1.

- Done

Table 1. It is suggested that authors make more use of the information in their table. It is only mentioned once in the main document.

- Done

Figure 2. Authors should explain why different green and red symbols are presented.  Besides, it is suggested that the figure be separated into agonists and antagonists. Furthermore, it is suggested that the agonists be placed in the order of their appearance in the main document. The way they are presented is confusing, and they do not have an order of appearance in the figure. In addition, the use of panels would be a better way of presentation.

- Figure 2 has been revised accordingly

2.1.3. Double-knot toxin (DkTx). For a better understanding of the readers, it is suggested that the information corresponding to figure 3 be indicated first and then the information corresponding to figure 4 be added. The way this paragraph is presented is confusing because Figure 4 is indicated first and then Figure 3.

- Change has been done accordingly

Lines 353-355. Please verify the following sentence: "Both sites are localized at extracellular positions (Figure 4) critical for the channel function, determined by cryo-EM of TRPV1 bound to DkTx [25]: " This information is not related to Figure 4. Please verify. Besides, it is suggested that this information be indicated in Figure 2.

- Information about the DkTx binding site to TRPV1 is shown in Figure 3. The change has been made in the text.

Figure 3. Please include the symbols with the corresponding names in the figure legends.

- Done

Lines 560-569. It is suggested that the authors introduce a scheme regarding this paragraph

- For greater clarity, this paragraph has been deleted (L563-566).

We sincerely thank Reviewer 1 for reviewing our article.

Reviewer 2 Report

Comments and Suggestions for Authors

The manuscript is well-written, engaging, and interesting. The clarity and flow of the text make it easy to read. Furthermore, the diagrams are both aesthetically pleasing and highly informative, significantly enhancing the overall presentation of the work.

I recommend this paper for publication. From a scientific point of view, I find the content to be rigorous, comprehensive, and without any notable deficiencies. No additional comments or revisions are necessary in this regard.

I have few editorial comments:

Line 66: There is only reference for Figure 1 and figure 1C, but there is no reference for Figure 1A and 1B. The reference for Fig 1A and 1B is shown in line 91. Maybe the  order of figures should be changes – first capsaicin and then structure of TRPV1? Please clarify that.

Line 80: pH5 – should it be pH=5. The same for line 82

Line 138: comma after function to be removed

Line 143: last name of the author instead of a number. There is no such author in the reference

Line 499: in should be with a capital letter

Line 557: last name of the author instead of a number.

Author Response

Line 66: There is only reference for Figure 1 and figure 1C, but there is no reference for Figure 1A and 1B. The reference for Fig 1A and 1B is shown in line 91. Maybe the  order of figures should be changes – first capsaicin and then structure of TRPV1? Please clarify that.

Changes have been made in section 1.1 to correct the recall in Figure 1.

Line 80: pH5 – should it be pH=5. The same for line 82

Done

Line 138: comma after function to be removed

Done

Line 143: last name of the author instead of a number. There is no such author in the reference

Reference has been removed from the text

Line 499: in should be with a capital letter

Done

Line 557: last name of the author instead of a number.

Done

We sincerely thank Reviewer 2 for reviewing our article.